# Comparison of the clinical effects of computer-assisted and traditional techniques in bilateral total knee arthroplasty: A meta-analysis of randomized controlled trials

**Liangjun Zhao[☯], Fang Xu[☯], Shan Lao, Jingmin Zhao, Qingjun Wei[ID]***

Department of Bone and Joint Surgery, The First Affiliated Hospital of Guangxi Medical University, Nanning, China

☯ These authors contributed equally to this work.
* weiqingjungxnn@163.com

**Data Availability Statement:** All relevant data are within the manuscript and its Supporting Information files.

## Abstract

### Background

It is unclear whether there are individual differences in the long-term efficacy of computer-assisted and traditional total knee arthroplasty. The purpose of this study was to perform a meta-analysis comparing the same individuals undergoing computer-assisted and traditional total knee arthroplasty separately to determine whether computer-assisted total knee arthroplasty can provide better lower extremity radiographic results and clinical outcomes.

### Methods

We searched literatures to identify relevant randomized controlled trials comparing the effects of computer-assisted and traditional methods in bilateral total knee arthroplasty. After screening, quality evaluation and data extraction according to inclusion and exclusion criteria, the quality and bias risks of the included studies were evaluated. The meta-analysis compared the radiographic results, functional outcomes and complications of the two techniques.

### Results

Six clinical controlled trials were included, with total of 1098 patients. The meta-analysis showed that the accuracy in terms of the mechanical axis of the lower extremity, the sagittal alignment of the femoral component and the coronal alignment of the tibial component in computer-assisted total knee arthroplasty was significantly better than those in traditional total knee arthroplasty. There were no differences in the functional results, revision rates or aseptic loosening rates between the two techniques.

### Conclusion

After excluding individual differences such as bone development and bone quality, although computer-assisted techniques can better accurately correct the mechanical axis of the lower extremity and the position of prosthesis implantation than traditional techniques, there

**Funding:** The author(s) received no specific funding for this work.

**Competing interests:** The authors have declared that no competing interests exist.

is no significant difference in the functional results and revision rate of bilateral total knee arthroplasty in the same individual.

## Introduction

The precise matching of the prosthetic component is closely related to the clinical effect in total knee arthroplasty (TKA) [1–3]. The ideal prosthetic implantation restores the mechanical axis of the lower extremity to the range of 0˚ ± 3˚ [1, 4–6]. Computer-assisted TKA has recently made remarkable developments in clinical application. Professors generally believed that computer-assisted TKA can effectively reduce the occurrence of misaligned outliers greater 3˚ [7–9]. However, it is also reported that there is no difference between computer-assisted TKA and traditional TKA in eliminating outliers [10, 11]. So, comparing with conventional TKA, whether computer-assisted TKA can improve radiographic results and functional outcomes has still been controversial [4, 12].

There are different opinions from randomized controlled tests (RCTs) on the efficacy of computer-assisted and traditional TKA [12–17], whether the reason for the different conclusions is that these two surgical techniques are separately carried out in different patients. Because of the differences in the bone development form of lower limbs, bone quality(affected by differences in age, sex, or body mass index) and personal subjective factors(may be affected knee functional scores) in different patients [12, 18, 19], the conclusion may be biased. We regard these differences as individual differences. Recently, the RCTs of computer-aided and traditional TKA in the same person reported [12, 20–22], which the aim is to reduce the adverse effect of individual differences on the conclusion. Published meta-analyses showed that the radiographic results and clinical efficacy of computer-assisted and conventional TKA did not include consideration of individual differences [17, 23] Whether individual differences lead to contradictory research results is not very clear from the current reports.

The purpose of this study was to perform a meta-analysis of RCTs on bilateral TKA in the same individual to compare radiographic results, knee function, and long-term prosthetic survival rates, assuming that individual differences such as systemic bone development and bone structure were excluded. Under this circumstance, we studied whether computer-assisted TKA has a higher prosthesis alignment rate and better clinical efficacy than conventional TKA.

## Methods

### Search strategy and study selection

This study was implemented following the guidelines of the Preferred Reporting Items for Systematic Reviews and Meta-Analyses (PRISMA) statement and was based on the Cochrane review methods. We searched the Embase, PubMed, Cochrane Library and Web of Science databases from database construction to December 31, 2019. Search keywords included "total knee arthroplasty OR total knee replacement OR TKA OR TKR", "computer OR assisted OR navigation OR navigated", and "bilateral". At the same time, relevant research articles were retrieved by searching the cited references and review articles.

The inclusion criteria were as follows: (1) included RCTs comparing the outcomes between computer-assisted TKA and conventional TKA; (2) patients underwent TKA for the first time; (3) the studies were available in English language; (4) comparison of radiographic results or functional outcomes of computer-assisted vs conventional TKA using at least 1 outcome

measure was done; (5) continuous variable indicators including sample size, mean, and standard deviation.

The exclusion criteria were as follows: (1) repeat publication of a study; (2) study for which raw data cannot be obtained; (4) case reports, conference materials, animal experiment studies, cadaveric mechanical tests; (5) retrospective studies.

Two reviewers independently read the title and abstract of the studies and selected eligible studies for full text review. Determining which articles to include required two reviewers to agree, disagreements over the literature selection were resolved by a third reviewer.

## Assessment of methodological quality

Cochrane risk of bias tools [24] were used by two reviewers to perform risk assessment of the RCTs. The outcomes of the quality assessment of each study required two reviewers to agree, and differences were resolved by a senior reviewer. The meta-analysis did not assess publication bias. When a meta-analysis includes at least 10 studies, funnel chart asymmetry tests are usually performed. Our meta-analysis included only 6 studies, so no asymmetry test was required.

## Data extraction

The basic information of the studies was extracted and summarized in an Excel table, including the first author, publication date, basic information of the study subjects (case number, age and sex), intervention measures, follow-up time, outcome indicators, etc. Among them, the outcome indicators included outliers with varus or valgus of the mechanical axis of the lower extremity $> 3°$, outliers with femoral and tibial prosthesis positioning deviation $> 3°$, the Knee Society Score (KSS), the Western Ontario and McMaster University Osteoarthritis Index (WOMAC) score, knee range of motion (ROM), complication rate and prosthetic survival rate. If the data related to the patients included in one study were unclear or missing, we emailed the corresponding authors for clarification. If the authors did not respond to the e-mail or if accurate data were not available, the meta-analysis excluded the data on such outcomes from the study.

## Statistical analysis

The binary classification variables were expressed by relative risk (RRs) and 95% confidence intervals (CIs). The continuous variables were expressed as the mean difference (MD) and 95% CI. The $I^2$ was calculated for heterogeneity, with 50% as a threshold for low or high heterogeneity. When $I^2 < 50\%$, a fixed effect model (FE) is used; if $I^2 \geq 50\%$, a random effect model (random effect, RE) is used. Forest plots were used to illustrate the results of each study, the pooled estimate of the effect, and the overall summary effect. Significance was set at $p < 0.05$. All statistical analyses were conducted using RevMan version 5.3.

## Results

### Flow of included studies

According to the search strategy, a total of 279 related studies were retrieved: 58 articles from Embase, 63 articles from PubMed, 85 articles from the Cochrane Library, and 73 articles from Web of Science. We identified 2 articles by manual search. We removed 189 duplicate studies. By reading the title and abstract, 55 irrelevant studies were excluded, and the remaining 37 related studies were initially screened. After further reading the full text and screening in strict

accordance with the inclusion and exclusion criteria (see above), we finally included 6 studies [13, 15, 21, 25–27]. The literature screening process and results are shown in Fig 1.

## Study characteristics and risk-of-bias assessment

Six studies included 1098 cases of computer-assisted TKA and 1098 cases of conventional TKA; we summarized basic information such as age, sex, and outcome indicators for all patients in Table 1. All studies contained radiographic data. A total of four studies [13, 15, 25, 27] included follow-up data after surgery, with an average follow-up of 1 to 15 years. There were three articles [13, 15, 27] all from the same author, but they had different patient groups, so they were included separately in this meta-analysis. Average age of subjects in one of the literatures was <60 years, so there may have been heterogeneity when comparing between groups. All studies followed these principles, which the radiologist, the surgeon who conducted the follow-up study, and the patients were all blinded with regard to the type of surgical procedure performed in each knee, also patients underwent bilateral total knee arthroplasty with one knee treated randomly with conventional total knee arthroplasty and the other treated with computer-assisted total knee arthroplasty. Five studies included that the bilateral procedure was performed sequentially during the same anesthetic session in each patient. Only one study [21] underwent staged bilateral TKAs within a period of 3 months.

Because the method of measuring the implantation angle was not uniform in each of the radiographic results, outliers that were more than 3° from the ideal angle were listed. Therefore, the data of outliers were included in this study. The clinical outcomes were also different. Among them, there were 3 reports of the KSS, 4 of the WOMAC score and 4 of ROM. 3 studies reported on complication rate which included infection, deep venous thrombosis of the lower limbs, periprosthetic fractures and aseptic loosening. 3 studies reported on aseptic loosening

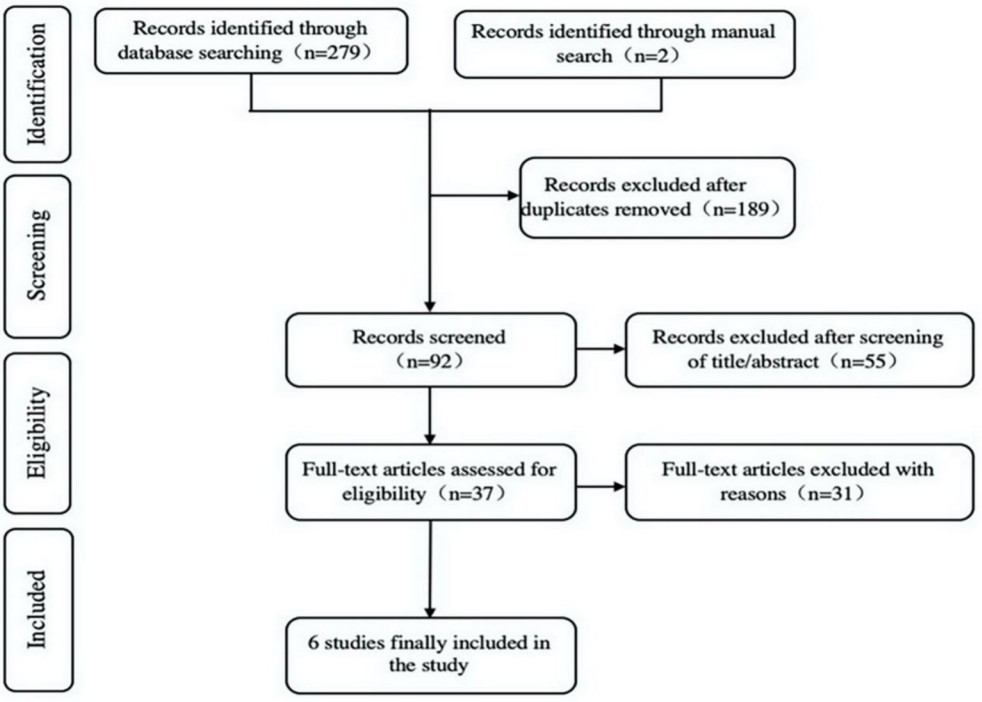

**Fig 1. Literature screening process and results.**

**Table 1. General information of included studies.**

| Study | Location of Study | Study design | Level of Evidence | Sample size | Age for surgery (Years) | Female/Male | Follow-up* (Years) | Outcomes | |
|---|---|---|---|---|---|---|---|---|---|
| | | | | | | | | Radiographic | Clinical |
| Zhang 2011 | China | RCT | 2 | 32 | 63 | 25/7 | NA | ①④ | NA |
| Kim 2017 | South Korea | RCT | 1 | 162 | 68.1 | 153/9 | 12.3 | ①②③④⑤ | ⑥⑦⑧⑨⑩ |
| Weng 2009 | Taiwan | RCT | 2 | 60 | 70 | 41/19 | NA | ①②③④⑤ | NA |
| Seon 2007 | South Korea | RCT | 2 | 42 | 64.2 | 33/9 | 1 | ①②④ | ⑥⑧ |
| Kim 2012 | South Korea | RCT | 1 | 520 | 68 | 452/68 | 11.1 | ①②③④⑤ | ⑥⑦⑧⑨⑩ |
| Kim 2018 | South Korea | RCT | 1 | 282 | 59 | 223/59 | 15 | ①②③④⑤ | ⑥⑦⑧⑨⑩ |

*Average follow-up time. NA = no data. Outcomes: ① Outliers with the force line of the lower limb in the coronal position more than 3˚ in entropion or valgus; ② Outliers with the deviation of the femoral prosthesis in the coronal plane more than 3˚; ③ Outliers with the deviation of sagittal femoral prosthesis implantation angle more than 3˚; ④Outliers with the deviation of coronal tibial prosthesis implantation angle more than 3˚; ⑤Outliers with the deviation of sagittal tibial prosthesis implantation angle more than 3˚; ⑥WOMAC; ⑦KSS; ⑧ROM; ⑨Complications; ⑩survival rate.

or revision rate. And 3 studies only considered revision as a result of aseptic loosening. Finally, a meta-analysis of the above indicators was performed. The results of the risk assessment for all RCTs are shown in Fig 2.

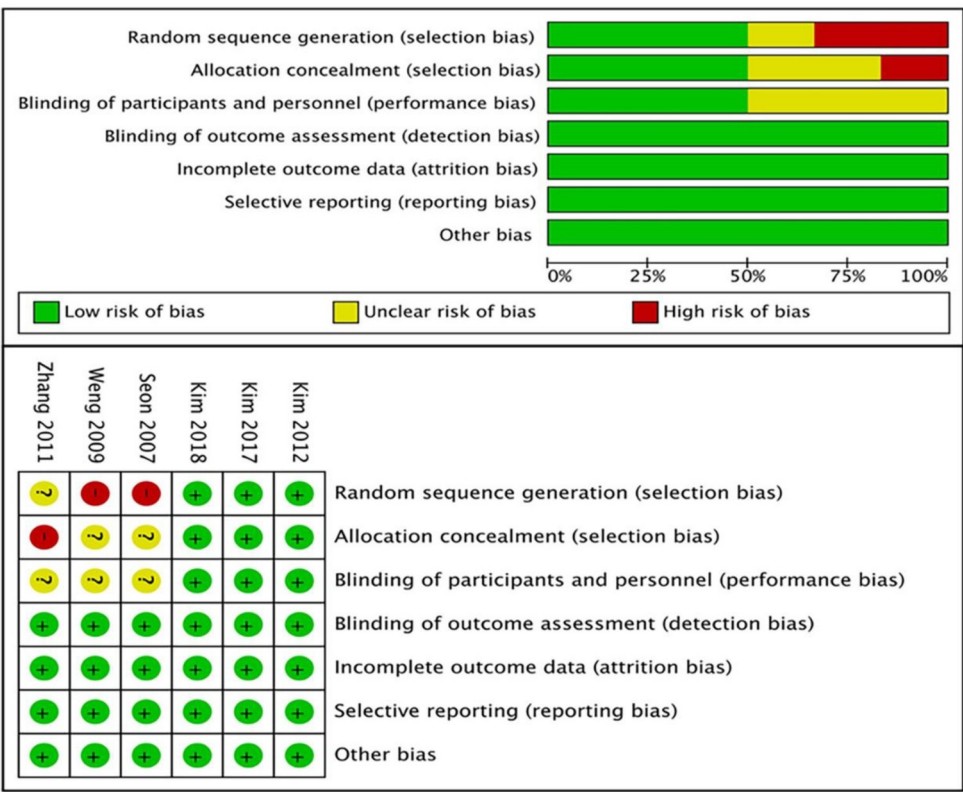

**Fig 2. Risk of bias graph.** "Plus" indicates a low risk of bias; "minus" indicates a high risk of bias; and "question mark" indicates unclear or unknown risk of bias.

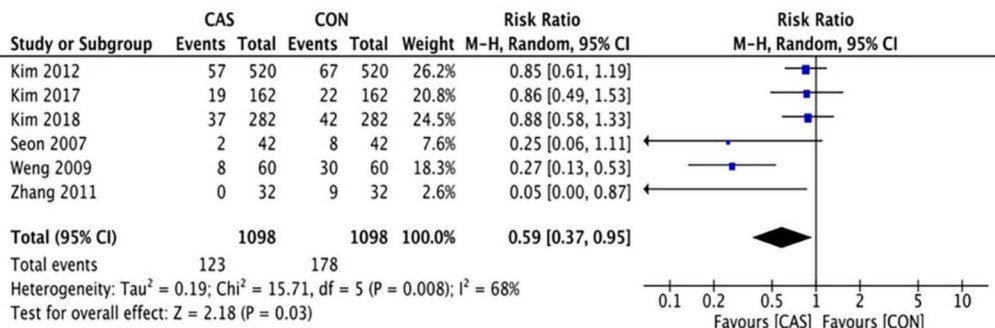

**Fig 3. Forest plot of the outlier rate of lower-extremity alignment before sensitivity analysis.**

## Radiographic results

A total of 6 articles were included in this analysis of the outliers of lower-limb alignment, which included 1098 cases of computer-assisted TKA and 1098 cases of conventional TKA. The outliers of the mechanical axis of the lower extremity included 123 cases in the CAS group and 178 cases in the CON group (RR = 0.59 [95% CI, 0.37 to 0.95]; P = 0.03; $I^2$ = 68%) (Fig 3), and the difference was statistically significant (P <0.05). The outliers of the implantation angle of the femoral prosthesis in the coronal plane included 98 cases in the CAS group and 116 cases in the CON group (RR = 0.84 [95% CI, 0.65 to 1.09]; P = 0.20; $I^2$ = 0%) (Fig 4), there were no significant differences between the two groups. The outliers of the femoral prosthesis implantation angle in the sagittal plane included 81 cases in the CAS group and 124 cases in the CON group (RR = 0.65 [95% CI, 0.51 to 0.84]; P = 0.001; $I^2$ = 0%) (Fig 5), the difference was statistically significant (P <0.05). The outliers of the tibial prosthesis implantation angle in the coronal plane included 109 cases in the CAS group and 149 cases in the CON group (RR = 0.66, [95% CI (0.51 to 0.86); P = 0.008; $I^2$ = 0%) (Fig 6), the difference was statistically significant (P <0.05). The outliers of the tibial prosthesis implantation angle in the sagittal plane included 253 cases in the CAS group and 266 cases in the CON group (RR = 0.95, [95% CI 0.84 to 1.07]; P = 0.41; $I^2$ = 11%) (Fig 7), there was no significant difference.

## Functional outcomes

Among this analysis, we performed the KSS, the WOMAC score, the ROM, and the complications after surgery. The KSS knee and the KSS function was separately evaluated in 3 studies

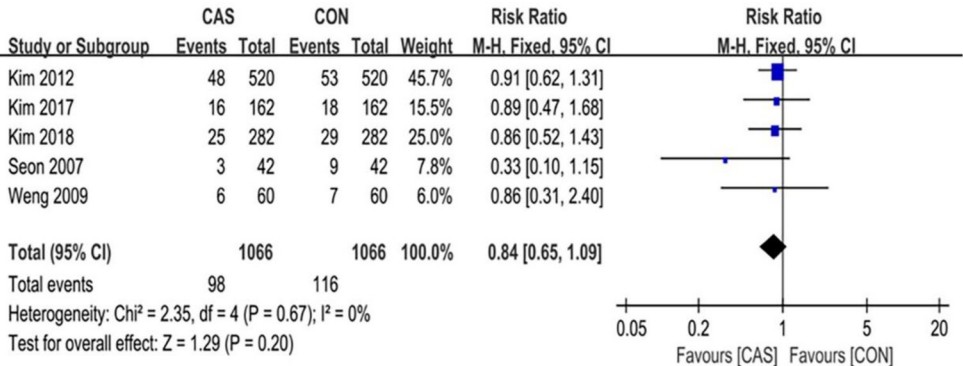

**Fig 4. Forest plot of the outlier rate of the coronal alignment of the femoral component.**

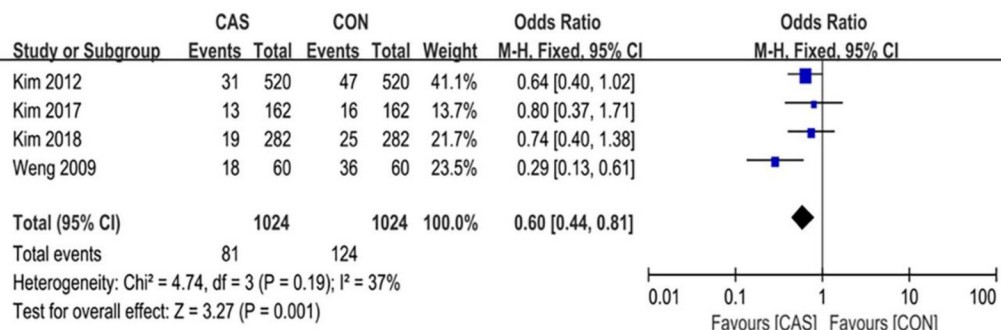

**Fig 5. Forest plot of the outlier rate of the sagittal alignment of the femoral component.**

with an average follow-up of over 8 years. The pooled mean difference in the KSS knee was 0.46 (95% CI, -0.63 to 1.56; P = 0.41; $I^2$ = 75%) and the pooled mean difference in the KSS function was -0.75 (95% CI, -2.21 to 0.71; P = 0.31; $I^2$ = 76%) (Fig 8). The WOMAC score was evaluated in 4 studies with an average follow-up of over 1 year, and the pooled mean difference was -0.42 (95% CI, -2.15 to 1.30; P = 0.63; $I^2$ = 81%) (Fig 9). The range of motion was evaluated in 4 studies with an average follow-up of over 1 year, and the pooled mean difference was -0.39 (95% CI, -2.08 to 1.30; P = 0.65; $I^2$ = 70%) (Fig 10). The mean differences in the KSS knee, the KSS function and the WOMAC score and the range of motion were not significant.

## Complication rate and survivorship

A total of 3 studies reported postoperative complications with an average follow-up of over 8 years, which included 14 cases in the CAS group and in 9 cases in the CON group (RR = 1.56 [95% CI, 0.67 to 3.63]; P = 0.30; $I^2$ = 0%) (Fig 11), there was no significant difference in the postoperative complication rate between the two groups. 3 studies reported revision rate due to aseptic loosening with an average follow-up of over 8 years, with 8 cases in the CAS group and 6 cases in the CON group (RR = 1.33 [95% CI, 0.46 to 3.83]; P = 0.59; $I^2$ = 0%) (Fig 12), and there was no significant difference.

## Discussion

The previous meta-analyses [17, 28–30] were performed to compare the efficacy of computer-assisted and traditional total knee arthroplasty in different two groups. Although literatures

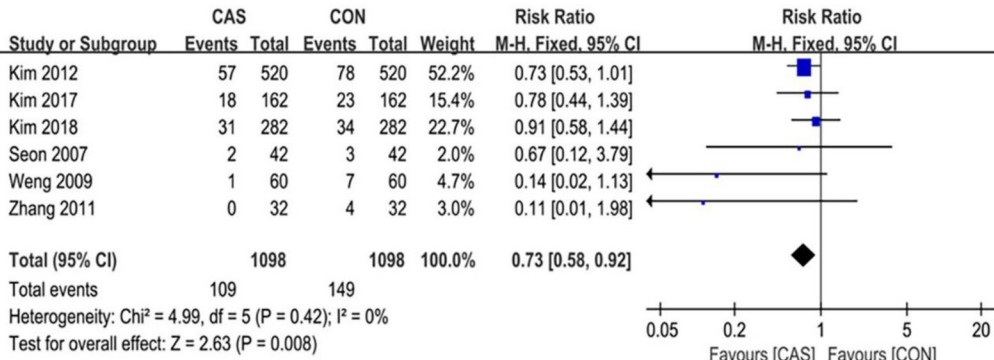

**Fig 6. Forest plot of the outlier rate of the coronal alignment of the tibial component.**

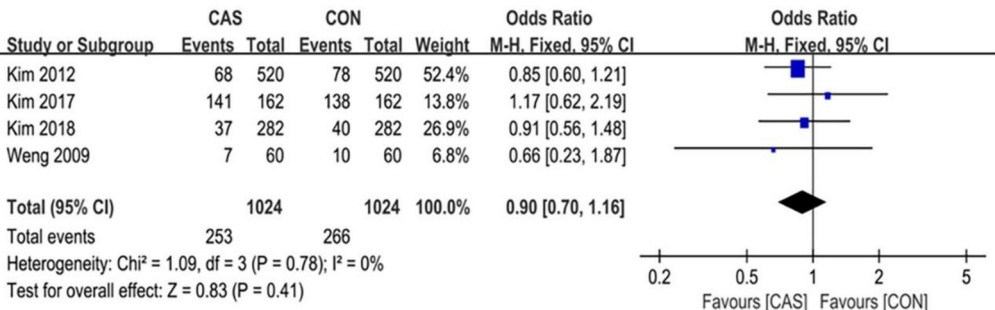

**Fig 7. Forest plot of the outlier rate of the sagittal alignment of the tibial component.**

which were included follow the principle of random control [31–36], there is still room for improvement in eliminating confounding variables. The confounding variables that have not been eliminated mainly include the differences of bone development form of lower limbs, osteoporosis degree, physical and mental condition in different patients [12, 20, 21, 37]. We consider that the former two variables may affect the effect of total knee replacement and the survival time of prosthesis to some extent. The last may lead to uncontrollable risk for subjective score of postoperative follow-up. Therefore, comparing computer-assisted and traditional total knee arthroplasty on both sides of the same person will eliminate the above confounding variables [13, 15, 21, 25–27]. So, it is necessary to use these studies for meta-analysis of radiographic results, functional results, and prosthetic survival.

Five studies included that the bilateral procedure was performed sequentially during the same anesthetic session in each patient. Only one study underwent staged bilateral TKAs within a period of 3 months. The choice of surgical treatment within a certain period can avoid the risk of uncertainty caused by individual differences such as body mass index, level of activity, knee deformity, and osteoporosis. Each patient was as self-control through bilateral knee joints, making the knee function outcomes more accurate and objective than using different individuals as control and avoiding the impact of physical and mental factors [12, 18, 19]. The objects included in the study had similar leg alignment (mechanical axis) before surgery on both sides (13.2±6.93 vs 13.1±5.84, p = 0.976). This similarity suggested that there was a correlation between knee deformities before surgery in the two groups, which cannot impact accurate evaluation of the orthopedic effect of the two techniques by dismissing the risk of different knee deformities [21].

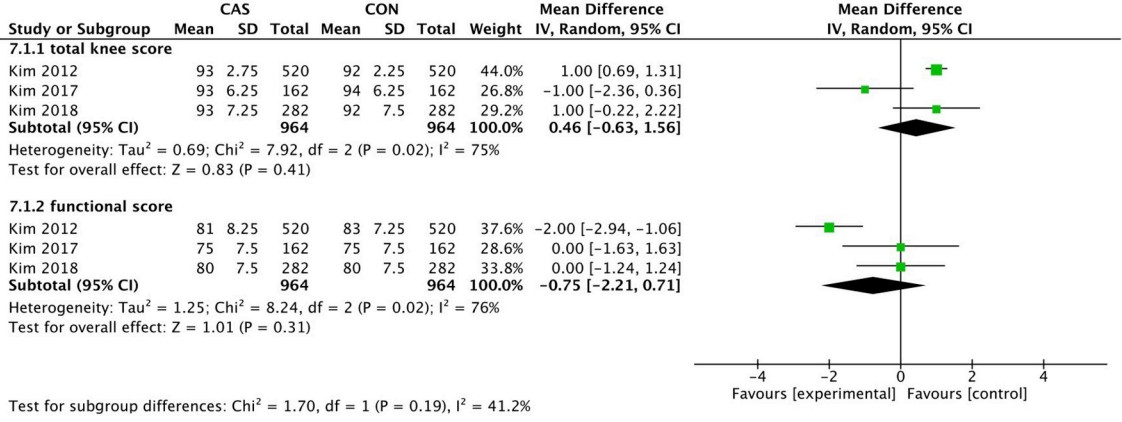

**Fig 8. Forest plot of the KSS.**

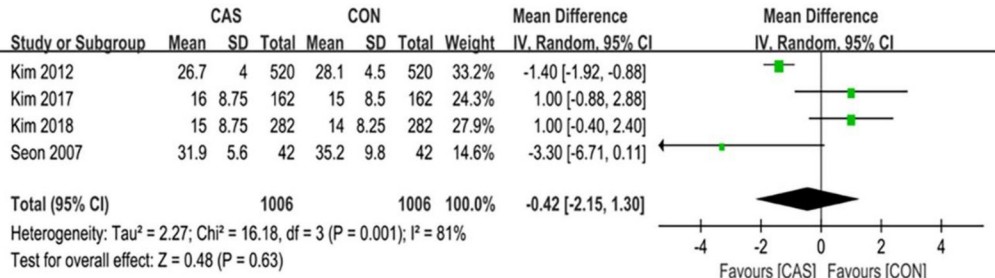

**Fig 9. Forest plot of the WOMAC scores.**

There are studies showing that when the deviation of the alignment angle between the mechanical axis and the prosthetic component is controlled within 3˚, knee function and prosthesis life is improved [6, 38, 39]. Before assessing the differences in long-term follow-up results for computer-assisted TKA and conventional TKA, it is necessary to evaluate whether there are individual differences between the two techniques in terms of knee force lines, prosthetic accuracy, and degree of matching. The restoration of the alignments of the lower limbs, especially the mechanical axis of the coronal plane, is the main factor affecting the long-term efficacy of TKA. Based on the assumption that poor alignment of the lower limbs or components after TKA will affect the survival of the prosthesis, a clinically introduced computer-assisted system can obtain a better prosthetic force line and matching degree than those in traditional techniques [21]. The radiographic indicators included in this meta-analysis contained 5 types of component alignment (the deviation rate of the mechanical axis from neutral alignment and the deviation rate of femoral and tibial prosthesis alignment in the coronal and sagittal planes). The results showed that the accuracy of the mechanical axis of the lower limb, the sagittal alignment of the femoral component and the coronal alignment of the tibial component was better in the computer-assisted group than in the conventional group, which was consistent with the published meta-analysis results. The component implantation in the computer assisted group appeared more accurate alignment, which compared with conventional group [17, 23, 40–43].

Although published meta-analysis results show that the computer-assisted group can effectively improve the alignment of prosthetic components, the results in the included literature [13, 15, 25, 27] showed that the two techniques achieved similar clinical effects, it was not enough to prove that the fit of the prosthesis was not related to knee function. In this meta-analysis, we evaluated the ROM, KSS, and WOMAC score, and the results showed that the mean difference in ROM between the two groups was 0.81˚. The differences in knee score and

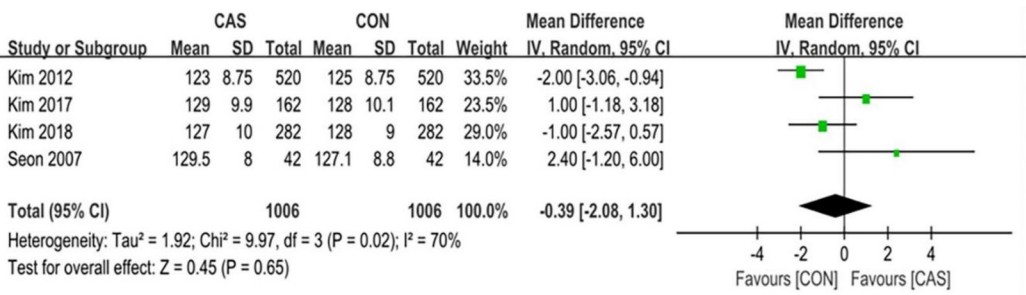

**Fig 10. Forest plot of the range of motion.**

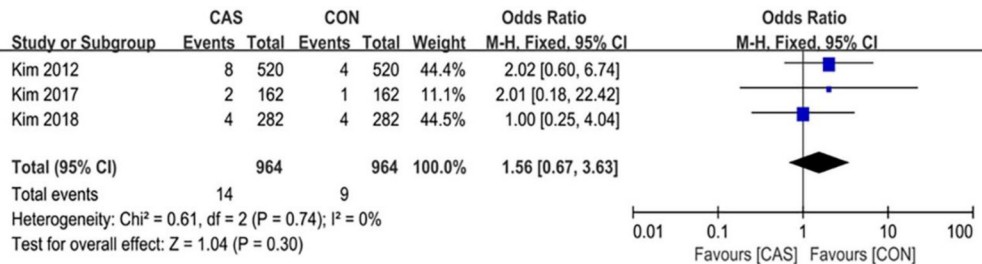

**Fig 11. Forest plot of the complication rate.**

functional score were 0.46 and 0.75 points, respectively, and the difference in WOMAC score was 0.51. The mean differences of several scores were small. We also found no significant difference in the incidence of postoperative indicators, such as surgical mouth infection, thrombosis of the lower limb, periprosthetic fractures, and aseptic loosening of the prosthesis. There are also studies reporting that computer-assisted TKA can restore better knee function and provide a better quality of life, which compared with conventional TKA [44]. However, more meta-analysis results show that there is no significant difference between computer-assisted TKA and conventional TKA [11, 20, 45–47]. In particular, it has been reported that computer-assisted TKA may achieve good clinical results in patients with severe knee deformities [13, 46]. In most studies, randomized controlled clinical trials were designed without considering the preoperative data of patients, such as the degree of bilateral lower limb or knee joint deformities, which may cause bias in the effect of recovery of the mechanical axis of the lower limb between the two groups. In this analysis, 1 article included the indicator of knee deformity before the operation, and 2 articles compared the perioperative indicators, such as the operation time and the amount of bleeding during the operation [21, 26]. The comparison of preoperative data between the two groups showed that the bleeding volume (mL) of the computer-assisted group was less than that of the conventional group (619 ± 268 mL vs. 736 ± 358 mL, P = 0.025). The mean tourniquet time (min) of the computer-assisted group was significantly longer than that of the conventional group (93.7 ± 22.7 min vs. 72.0 ± 20.5 min, P <0.0001). There was no significant difference in the mean length of hospital stay (days, d) between the two groups (6.63 ± 1.44 d vs. 6.59 ± 1.66 d, P = 0.150). Although it caused longer operation time but less amount bleeding in the computer-assisted group. It also did not increase the length of hospitalization and rehabilitation after surgery in the computer-assisted group. Moreover, in the kinematic alignment group, the precise alignment of the knee prosthesis, anatomical matching, and maintenance of a good medial and lateral space balance were positively correlated with postoperative functional results [48, 49]. Computer-assisted TKA can

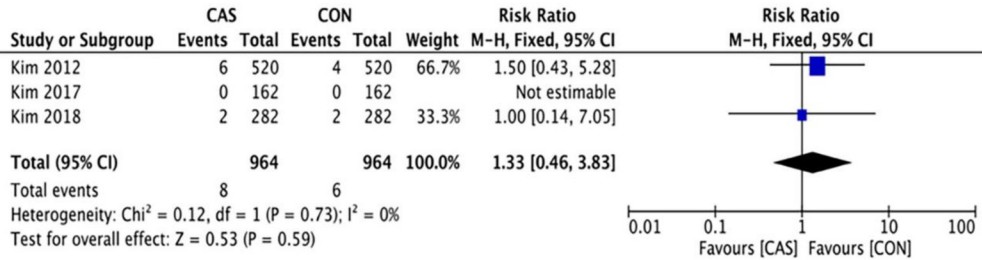

**Fig 12. Forest plot of the revision rate.**

make full use of the advantages of computer navigation for accurate osteotomy and suitable body matching for surgical treatment of patients with severe knee deformities, especially for patients with extra-articular deformities and severe femoral arch flexion [13, 46]. All studies in this study selected mechanical alignment as the alignment standard. If kinematic alignment was used, different results were obtained. The number of included studies is small, and there is not a large amount of data for analysis. More RCTs are needed for further analysis.

Our study also has several limitations. (1) Six studies were included in this meta-analysis, and the radiographic parameters used to evaluate alignment in the studies were not uniform. Therefore, we evaluated the outliers to assess the accuracy of the two techniques. (2) Although there are also some literatures of the same author in this meta-analysis, these research data are not repeated, so the possibility of influencing the conclusion is very small. (3) Some risk factors for specific surgical techniques, the computer navigation systems, selection of the implant types and operation designs are different in each literature of this study. (4) In this study, the time span of the literature was slightly larger, and the study areas were different. Some authors may have a better-than-average surgical environment, which may cause the accuracy of prosthesis implantation and the postoperative curative effect to be quite different. (5) Four articles containing clinical outcomes also had varying follow-up times. Finally, the results of this study cannot be considered conclusive. There are still unclear relationships between alignment and long-term clinical outcomes indicating that more studies are needed to assess these relationships.

## Conclusion

In summary, although we observed better postoperative prosthesis alignment results in the computer-assisted group, we did not observe significant differences in long-term functional outcomes and prosthetic loosening rates between two techniques, after ruling out influential factors caused by individual differences. Therefore, when evaluating the application of computer navigation in total knee arthroplasty, it is necessary to include more prospective randomized controlled trials to evaluate the practicality of computer navigation technology in knee replacement surgery.

## Supporting information

**S1 Checklist. PRISMA 2009 checklist.**
(DOC)

**S1 Data. Data of the study.**
(XLSX)

## Acknowledgments

We would like to thank the staff of the Department of Orthopedic Surgery, First Affiliated Hospital of Guangxi Medical University, for their support and American Journal Experts for their assistance with the language of the manuscript.

## Author Contributions

**Conceptualization:** Liangjun Zhao.

**Data curation:** Liangjun Zhao.

**Formal analysis:** Qingjun Wei.

**Funding acquisition:** Qingjun Wei.

**Project administration:** Shan Lao.

**Software:** Jingmin Zhao.

**Supervision:** Fang Xu.

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
