## [Decision Letter · Decision Letter 0]

8 May 2020

PONE-D-20-07787

Comparison of the clinical effects of computer-assisted and traditional techniques in bilateral total knee arthroplasty: a meta-analysis of randomized controlled trials

PLOS ONE

Dear Mr Wei,

Thank you for submitting your manuscript to PLOS ONE. After careful consideration, we feel that it has merit but does not fully meet PLOS ONE’s publication criteria as it currently stands. Therefore, we invite you to submit a revised version of the manuscript that addresses the points raised during the review process.

We would appreciate receiving your revised manuscript by Jun 22 2020 11:59PM. To enhance the reproducibility of your results, we recommend that if applicable you deposit your laboratory protocols in protocols.io, where a protocol can be assigned its own identifier (DOI) such that it can be cited independently in the future. For instructions see: http://journals.plos.org/plosone/s/submission-guidelines#loc-laboratory-protocols

We look forward to receiving your revised manuscript.

Kind regards,

Osama Farouk

Academic Editor

PLOS ONE

Journal Requirements:

1.

3, Please include captions for your Supporting Information files at the end of your manuscript, and update any in-text citations to match accordingly. Please see our Supporting Information guidelines for more information: http://journals.plos.org/plosone/s/supporting-information.

Reviewers' comments:

Reviewer's Responses to Questions

**Comments to the Author**

1. Is the manuscript technically sound, and do the data support the conclusions?

Reviewer #1: Partly

Reviewer #2: Partly

2. Has the statistical analysis been performed appropriately and rigorously? 

Reviewer #1: I Don't Know

Reviewer #2: Yes

3. Have the authors made all data underlying the findings in their manuscript fully available?

Reviewer #1: Yes

Reviewer #2: Yes

4. Is the manuscript presented in an intelligible fashion and written in standard English?

Reviewer #1: Yes

Reviewer #2: Yes

5. Review Comments to the Author

Reviewer #1: This article adequately summarizes the literature using this kind of meta-analysis of RCTs with self-control to decrease the subjectivity between the participants. This article emphasizes on the importance of doing more RCTs concerning the comparison between the CAOS and the conventional technique

• You used the term bilateral in the title and it was not clear to me if it is bilateral simultaneous or staged

• How would you assess the bone development and bone quality issues before excluding these patients?

• I believe the authors should have included other articles from other languages (selection bias)

• I believe a table of patient characteristics and the differences between them with providing the P-value would be of extreme importance here to properly interpret the results presented in this study.

• In the conclusion section, please rephrase the first sentence because it is too long that I did not get the idea behind it

• In the discussion section, page 17, first paragraph: please correct the word “coronary alignment”

Reviewer #2: The concept of a meta-analysis of bilateral TKA in which one side underwent computer navigation assisted surgery while the other side underwent conventional TKA is interesting, and may help clarify whether the navigation's extra cost, time, and training are justified. Generally, this entire study can be shortened and focused to emphasize the point rather than getting overcomplicated with the finer details. This begins in the abstract itself, which contains too much information especially in the methods, results, and conclusion. The methods section can simply state that a review of the literature was performed (timing and exact sources unnecessary until later description of the study), and the type of software used is unnecessary here. In the results portion of the abstract, a simple summary of the findings can focus the reader, including which findings were significant and which were not (statistical information not necessary here). The conclusion should concisely state in a sentence or 2 the overall finding that computer-assisted surgery can improve the mechanical axis, though there may not be a significant difference in functional and revision results.

The body of the manuscript is very well researched, with appropriate supporting citations. It may be necessary to include a citation after the first sentence of the paper, as many - especially those against the use of navigation assisted surgery - can argue excellent results without "precise" implant matching in the absence of a reference.

Though the manuscript should have line numbers for reference of specific sentences during the review process, attempts will be made to locate precise concerns:

- First paragraph: are "CAS" and "CON" accepted terminology for these procedures? The need for abbreviation is understood, but it can get confusing later in the paper when non-universal terms are used repeatedly

- The end 3 sentences of first paragraph are confusing and perhaps mixing the same idea as if different. It should be emphasized that the primary positive outcome of computer assisted navigation is to eliminate outliers, not to improve "accuracy." One sentence with the appropriate references would make this section easier to read.

-Beginning of 2nd paragraph is confusing, due to the use of the term "individual differences". It is true that most studies do not compare navigation TKA vs conventional TKA within the same person, however these are large volume metaanalyses including RCAs that are cited, in which the entire point of the study design is to eliminate confounding variables through patient matching, and thus address "individual differences" as best as possible. This needs to be addressed in a way that acknowledges the need to eliminate confounding variables that does not appear to dismiss them as inadequate.

- The methods section is too long and wordy. Assume the reader knows how a search was done (don't need to include things like "we used computers." Tell the reader which databases were used and what keywords were searched.

- In patient selection: "patients underwent TKA for the first time". If the age, sex, and ethnicity weren't addressed, don't include this, it simply makes the paragraph longer and harder to read.

- The inclusion criteria seems to only include studies that included outliers greater than 3 degrees outside the target zone. This introduces a bias into the study. If a large scale randomized control trial was performed and showed equivalent outcomes with no outliers using either method of TKA, was this study then excluded? This is not included in the exclusion criteria.

- Your exclusion criteria includes studies with nonclinical results and incomplete measurement results. Then under data extraction there is a section about contacting authors and excluding study portions if data was unattainable. If these studies were not included, this entire paragraph under data extraction is unnecessary. If these studies were included but data was missing it seems they should have fallen into the exclusion criteria. This is confusing.

- The section about I-squared can be condensed into one short sentence, saying something along the lines of: "I squared was calculated for heterogeneity, with 50% as a threshold for low or high heterogeneity."

- The data section should simply present the results. There is too much analysis here about how they were obtained though this should occur in the discussion section. This section can be condensed for each outcome into one sentence stating the statistical results of each group. Alternatively, as the results are the same in word and table form, the very nice tables in each section can serve as the entire results section. The rest is unnecessary and confuses the reader

- For the KSS, WOMAC, ROM, complication rate, and aseptic loosening sections, a time period is necessary. No mention is given to the differences of when each of the 5-6 studies included found these results.

- Again the beginning of the discussion ignores the concept of randomized control groups matching patients in prior studies to eliminate confounding variables. Your study may improve by matching knees in the same individual but you have to acknowledge the ability of prior studies to have been conducted in an appropriate manner too.

- You also need to address the risk of performing surgery differently on the same patient. This may introduce a performance bias, as a patient who had a conventional TKA in the past knows s/he is having navigation assisted surgery on the 2nd knee and may perform better to improve the functional results. Were the studies you looked at randomized so that some patients received navigation assisted TKA first and then conventional 2nd? This information is necessary.

- The entire discussion of clinical results and correlation requires time periods to be addressed. It is mentioned that some of the studies were carried out over longer time than other studies but exact numbers are not provided.

Overall, given the issues noted above, including that there were only 6 studies (3 of which came from the same author), I'm not sure this truly qualifies as a metaanalysis. It is unclear how many patients were evaluated and over what time period. While a bit of interesting information regarding patients who have had both TKAs performed by a different method, it does not provide any new information in a way that is applicable to clinical practice. Less time should be focused on the statistical methods of obtaining this data and more on the meaningfulness of the findings and how to apply them to practice.

6. PLOS authors have the option to publish the peer review history of their article (what does this mean?). If published, this will include your full peer review and any attached files.

Reviewer #1: Yes: Mahmoud Hafez

Reviewer #2: No

---

## [Author Response · Author response to Decision Letter 0]

29 May 2020

Reviewer #1: This article adequately summarizes the literature using this kind of meta-analysis of RCTs with self-control to decrease the subjectivity between the participants. This article emphasizes on the importance of doing more RCTs concerning the comparison between the CAOS and the conventional technique

• You used the term bilateral in the title and it was not clear to me if it is bilateral simultaneous or staged

Five studies included that the bilateral procedure was performed sequentially during the same anesthetic session in each patient. Only one study underwent staged bilateral TKAs within a period of 3 months.

• How would you assess the bone development and bone quality issues before excluding these patients?

We think that osteoarthritis is a kind of degenerative joint disease caused by multiple factors. The main pathological manifestations are not inflammatory degeneration of articular cartilage and osteophyte formation at the joint edge. In the literatures, the author has compared the preoperative indexes of the two groups of patients. It is believed that the differences of bone morphology and bone density of bilateral knee joints can be reduced to the minimum in the same individual. Of course, the above differences are not completely excluded.

• I believe the authors should have included other articles from other languages (selection bias)

This study may be more comprehensive when it is included in the mata-analysis of other languages. However, at present, English has become one of the international languages, and almost all international authoritative magazines are published in English, indicating that English literature may represent the universality and popularity of research papers. Of course, we also consider that the future research will appropriately include multiple languages.

• I believe a table of patient characteristics and the differences between them with providing the P-value would be of extreme importance here to properly interpret the results presented in this study.

Because all of the data from six literatures of this meta-analysis were compared with computer-assisted and traditional techniques in bilateral total knee arthroplasty of the same individual, there was no difference in the characteristics of patients.

• In the conclusion section, please rephrase the first sentence because it is too long that I did not get the idea behind it

The first sentence of the conclusion has been restated.

• In the discussion section, page 17, first paragraph: please correct the word “coronary alignment”

The word "coronary alignment" has been corrected in the discussion section.

At all, your suggestion is very beneficial. I have revised it according to your suggestion. Thank you very much for reviewing this manuscript.

Reviewer #2: The concept of a meta-analysis of bilateral TKA in which one side underwent computer navigation assisted surgery while the other side underwent conventional TKA is interesting, and may help clarify whether the navigation's extra cost, time, and training are justified. Generally, this entire study can be shortened and focused to emphasize the point rather than getting overcomplicated with the finer details. This begins in the abstract itself, which contains too much information especially in the methods, results, and conclusion. The methods section can simply state that a review of the literature was performed (timing and exact sources unnecessary until later description of the study), and the type of software used is unnecessary here. In the results portion of the abstract, a simple summary of the findings can focus the reader, including which findings were significant and which were not (statistical information not necessary here). The conclusion should concisely state in a sentence or 2 the overall finding that computer-assisted surgery can improve the mechanical axis, though there may not be a significant difference in functional and revision results.

The summary has been simplified according your suggestion.

The body of the manuscript is very well researched, with appropriate supporting citations. It may be necessary to include a citation after the first sentence of the paper, as many - especially those against the use of navigation assisted surgery - can argue excellent results without "precise" implant matching in the absence of a reference.

We have already made a citation after the first sentence of the paper and adjusted the logic between sentences.

Though the manuscript should have line numbers for reference of specific sentences during the review process, attempts will be made to locate precise concerns:

- First paragraph: are "CAS" and "CON" accepted terminology for these procedures? The need for abbreviation is understood, but it can get confusing later in the paper when non-universal terms are used repeatedly.

"CAS" and "CON" have been revised to the full name, only the abbreviations in the pictures and tables are retained.

- The end 3 sentences of first paragraph are confusing and perhaps mixing the same idea as if different. It should be emphasized that the primary positive outcome of computer assisted navigation is to eliminate outliers, not to improve "accuracy." One sentence with the appropriate references would make this section easier to read.

The last three sentences of the first paragraph have been revised with appropriate references.

-Beginning of 2nd paragraph is confusing, due to the use of the term "individual differences". It is true that most studies do not compare navigation TKA vs conventional TKA within the same person, however these are large volume meta-analyses including RCAs that are cited, in which the entire point of the study design is to eliminate confounding variables through patient matching, and thus address "individual differences" as best as possible. This needs to be addressed in a way that acknowledges the need to eliminate confounding variables that does not appear to dismiss them as inadequate.

The second "individual differences" refers to the difference in bone development and bone of knee joint of patients. We think that the difference in the same individual can reduce the abnormal value, which has been re described.

- The methods section is too long and wordy. Assume the reader knows how a search was done (don't need to include things like "we used computers." Tell the reader which databases were used and what keywords were searched.

The methods section has been simplified again.

- In patient selection: "patients underwent TKA for the first time". If the age, sex, and ethnicity weren't addressed, don't include this, it simply makes the paragraph longer and harder to read.

The sentence has been restated.

- The inclusion criteria seems to only include studies that included outliers greater than 3 degrees outside the target zone. This introduces a bias into the study. If a large scale randomized control trial was performed and showed equivalent outcomes with no outliers using either method of TKA, was this study then excluded? This is not included in the exclusion criteria.

Outliers greater than 3 degrees is a recognized standard of prosthesis alignment anomaly in this field, and all relevant literatures we searched contain outliers, which it contains excluded literatures, so this cannot introduce a bias into the study.

- Your exclusion criteria includes studies with nonclinical results and incomplete measurement results. Then under data extraction there is a section about contacting authors and excluding study portions if data was unattainable. If these studies were not included, this entire paragraph under data extraction is unnecessary. If these studies were included but data was missing it seems they should have fallen into the exclusion criteria. This is confusing.

The exclusion criterion has eliminated the criterion of incomplete measurement results.

- The section about I-squared can be condensed into one short sentence, saying something along the lines of: "I squared was calculated for heterogeneity, with 50% as a threshold for low or high heterogeneity."

This section has been restated.

- The data section should simply present the results. There is too much analysis here about how they were obtained though this should occur in the discussion section. This section can be condensed for each outcome into one sentence stating the statistical results of each group. Alternatively, as the results are the same in word and table form, the very nice tables in each section can serve as the entire results section. The rest is unnecessary and confuses the reader.

The data section has been reorganized and expressed mainly to illustrate the results.

- For the KSS, WOMAC, ROM, complication rate, and aseptic loosening sections, a time period is necessary. No mention is given to the differences of when each of the 5-6 studies included found these results.

Follow-up time has been added to KSS, WOMAC, ROM, complication rate and aseptic loosening, which has been restated.

- Again the beginning of the discussion ignores the concept of randomized control groups matching patients in prior studies to eliminate confounding variables. Your study may improve by matching knees in the same individual but you have to acknowledge the ability of prior studies to have been conducted in an appropriate manner too.

In the published literature, the way to eliminate confounding variables is also reasonable. Our research is to perform surgical treatment on both sides of the knee joint of the same person, so as to compare and observe the postoperative results more effectively.

- You also need to address the risk of performing surgery differently on the same patient. This may introduce a performance bias, as a patient who had a conventional TKA in the past knows s/he is having navigation assisted surgery on the 2nd knee and may perform better to improve the functional results. Were the studies you looked at randomized so that some patients received navigation assisted TKA first and then conventional 2nd? This information is necessary.

All studies followed these principles，which the radiologist, the surgeon who conducted the follow-up study, and the patients were all blinded with regard to the type of surgical procedure performed in each knee, also patients underwent bilateral total knee arthroplasty with one knee treated randomly with conventional total knee arthroplasty and the other treated with computer-assisted total knee arthroplasty. Five studies included that the bilateral procedure was performed sequentially during the same anesthetic session in each patient. Only one study underwent staged bilateral TKAs within a period of 3 months. The above information has been restated in the paper.

- The entire discussion of clinical results and correlation requires time periods to be addressed. It is mentioned that some of the studies were carried out over longer time than other studies but exact numbers are not provided.

The exact number of follow-up time has been added. The follow-up time of the previous literature research has short-term and medium-term. Usually, the author makes statistics of the results of the last follow-up. Of course, as time goes on, more literature will be reported, and whether the results will change needs our continuous attention.

Overall, given the issues noted above, including that there were only 6 studies (3 of which came from the same author), I'm not sure this truly qualifies as a metaanalysis. It is unclear how many patients were evaluated and over what time period. While a bit of interesting information regarding patients who have had both TKAs performed by a different method, it does not provide any new information in a way that is applicable to clinical practice. Less time should be focused on the statistical methods of obtaining this data and more on the meaningfulness of the findings and how to apply them to practice.

Surely, there were three articles all from the same author, but they had different patient groups, so these data qualify as a meta-analysis. We have already restated the exact number of patients and time period. When we read the literature, we found that different authors reported that the postoperative results of computer-guided and conventional knee arthroplasty were inconsistent. Whether the results of computer-guided TKA were better than those of conventional TKA is still controversial. Because the inclusion criteria of the authors of the literature are inconsistent, it may lead to different results. We are considering whether the two groups of patients included in the literature are included individual differences such as knee joint development shape and bone quality have influenced the postoperative results. Therefore, we set the inclusion standard as a randomized controlled study of bilateral knee joint replacement of the same individual, so as to eliminate the abnormal values caused by individual factors as much as possible, and to observe the postoperative effects of both techniques objectively. Due to the limitation of bilateral knee joint replacement of the same individual, the reported literature is relatively limited, Of course, we believe that there will be more and more reports of this type of randomized controlled study, which needs further study in the future.

Finally, your suggestion is very beneficial. I have revised it according to your suggestion. Thank you very much for reviewing this manuscript.

We are very grateful to you for your valuable suggestions in your busy schedule, which greatly benefited for this paper. We have given your feedback carefully.

---

## [Decision Letter · Decision Letter 1]

25 Jun 2020

PONE-D-20-07787R1

Comparison of the clinical effects of computer-assisted and traditional techniques in bilateral total knee arthroplasty: a meta-analysis of randomized controlled trials

PLOS ONE

Dear Dr. Wei,

Thank you for submitting your manuscript to PLOS ONE. After careful consideration, we feel that it has merit but does not fully meet PLOS ONE’s publication criteria as it currently stands. Therefore, we invite you to submit a revised version of the manuscript that addresses the points raised during the review process.

We look forward to receiving your revised manuscript.

Kind regards,

Osama Farouk

Academic Editor

PLOS ONE

Reviewers' comments:

Reviewer's Responses to Questions

**Comments to the Author**

1. If the authors have adequately addressed your comments raised in a previous round of review and you feel that this manuscript is now acceptable for publication, you may indicate that here to bypass the “Comments to the Author” section, enter your conflict of interest statement in the “Confidential to Editor” section, and submit your "Accept" recommendation.

Reviewer #1: All comments have been addressed

Reviewer #3: All comments have been addressed

2. Is the manuscript technically sound, and do the data support the conclusions?

Reviewer #1: Yes

Reviewer #3: Yes

3. Has the statistical analysis been performed appropriately and rigorously? 

Reviewer #1: Yes

Reviewer #3: Yes

4. Have the authors made all data underlying the findings in their manuscript fully available?

Reviewer #1: Yes

Reviewer #3: Yes

5. Is the manuscript presented in an intelligible fashion and written in standard English?

Reviewer #1: Yes

Reviewer #3: No

6. Review Comments to the Author

Reviewer #1: Thank you for your great effort reviewing our comments.

I believe the authors have properly answered all my concerns and the manuscript is ready for publication.

Reviewer #3: Thank you for sending this paper for reviewing. There were some comments in the following:

1. This article has been revised, yet, the English writing style is still required to be polished a little to make it more readable.

2. Fig 1: The details of reasons for full-text articles exclusion should be provided.

3. P14: Only one study underwent staged bilateral TKAs within a period of 3 months Which one? Please indicate.

4. The conclusion is too long, and can be made to be shorten and concise.

7. PLOS authors have the option to publish the peer review history of their article (what does this mean?). If published, this will include your full peer review and any attached files.

Reviewer #1: **Yes: **Mahmoud A Hafez

Reviewer #3: **Yes: **Robert Wen-Wei Hsu

---

## [Author Response · Author response to Decision Letter 1]

12 Aug 2020

1.This article has been revised, yet, the English writing style is still required to be polished a little to make it more readable.

We read the full text carefully again and revised some places.

2. Fig 1: The details of reasons for full-text articles exclusion should be provided.

The exclusion criteria in this paper have been explained

3. P14: Only one study underwent staged bilateral TKAs within a period of 3 months. Which one? Please indicate.

We have already indicated it in this paper.

4. The conclusion is too long, and can be made to be shorten and concise.

We have simplified the conclusion part.

Thank you for your valuable comments！

---

## [Decision Letter · Decision Letter 2]

4 Sep 2020

Comparison of the clinical effects of computer-assisted and traditional techniques in bilateral total knee arthroplasty: a meta-analysis of randomized controlled trials

PONE-D-20-07787R2

Dear Dr. Wei,

We’re pleased to inform you that your manuscript has been judged scientifically suitable for publication and will be formally accepted for publication once it meets all outstanding technical requirements.

Kind regards,

Osama Farouk

Academic Editor

PLOS ONE

Additional Editor Comments (optional):

Reviewers' comments:

Reviewer's Responses to Questions

**Comments to the Author**

1. If the authors have adequately addressed your comments raised in a previous round of review and you feel that this manuscript is now acceptable for publication, you may indicate that here to bypass the “Comments to the Author” section, enter your conflict of interest statement in the “Confidential to Editor” section, and submit your "Accept" recommendation.

Reviewer #3: All comments have been addressed

2. Is the manuscript technically sound, and do the data support the conclusions?

Reviewer #3: Yes

3. Has the statistical analysis been performed appropriately and rigorously? 

Reviewer #3: Yes

4. Have the authors made all data underlying the findings in their manuscript fully available?

Reviewer #3: Yes

5. Is the manuscript presented in an intelligible fashion and written in standard English?

Reviewer #3: Yes

6. Review Comments to the Author

Reviewer #3: Dear Author,

After revision, the paper became readable. I suggest that the term “Traditional” might be changed to be “conventional”. The terminology of “conventional” is usually used in majority of literature. Otherwise, I had no further comments on this article.

7. PLOS authors have the option to publish the peer review history of their article (what does this mean?). If published, this will include your full peer review and any attached files.

Reviewer #3: **Yes: **Robert Wen-Wei Hsu

---

## [Editor Report · Acceptance letter]

9 Sep 2020

PONE-D-20-07787R2 

Comparison of the clinical effects of computer-assisted and traditional techniques in bilateral total knee arthroplasty: a meta-analysis of randomized controlled trials 

Dear Dr. Wei:

I'm pleased to inform you that your manuscript has been deemed suitable for publication in PLOS ONE. Congratulations! Your manuscript is now with our production department. 

Kind regards, 

on behalf of

Dr. Osama Farouk 

Academic Editor

PLOS ONE